# Engaging Children in Music-Making: A Feasibility Study Using Disabled Musicians as Mentors in Primary Schools

Eamonn McCarron [1], Erica Curran [1], Paul McQueen [2] and Roy McConkey [3,*]

1    Liberty Consortium, Derry BT48 7RE, UK
2    School of Education, Ulster University, Coleraine BT52 1SA, UK
3    Institute of Nursing and Health Research, Ulster University, Belfast BT15 1AP, UK
*    Correspondence: r.mcconkey@ulster.ac.uk

**Abstract:** The place of music in the school curriculum is under increasing threat, yet this is contrary to the growing evidence base of music's value to pupils' social and psychological development. A contributory factor is that many teachers report a lack of training, skill, confidence, or resources for excluding music in their classroom. An innovative project used young adults with disabilities as music mentors to improve children's access to creative music-making, while also providing non-threatening models to the children's teachers and providing them practical tools to embed music in their classrooms. An evaluation was undertaken of the project's impact. Eight classes from four schools in a city with high levels of social deprivation received 16 hours of music sessions over a four-week period, usually in the form of a four-hour workshop. Information was gathered from 171 children, the parents of 54 pupils; 8 class teachers; and 2 head teachers. The pupils' knowledge and appreciation of music showed significant improvements and their enthusiasm for music making had increased, which parents also confirmed. The teachers and head teachers identified five ways in which they had gained from Project Sparks with an increased appreciation of the potential value of music in the school curriculum. Further research is needed to identify how innovative projects can be sustained and extended to many more teachers and schools.

**Keywords:** music; primary schools; mentors; disability; intervention; feasibility study

## 1. Introduction

The place of music in the curriculum of UK schools is a continuing controversy.

Although music is designated a core subject, music programmes within primary as well as secondary schools are being reduced, marginalised, and even eliminated [1]. Similar declines have been reported in the US and other countries [2]. Despite classroom music maintaining its statutory position in UK national curricula, the common reasons for its decline include: a focus by schools on aspects of the curriculum for which they are most accountable (i.e., literacy, numeracy, and sciences) [3]; a lack of musical and pedagogical confidence among primary school teachers [4]; and a reduction in time allocation due to the inclusion of additional curriculum subjects such as those associated with Learning for Life and Work [5].

Paradoxically, this decline goes against a growing evidence base of the value of music to pupils' social and psychological development. A recent review identified three domains: as a catalyst for relationships, as means for self-expression and self-regulation, and a resource for self-transformation [6]. These arguments are important, but they are subsidiary to the fundamental principle and basic human right of every child's access to music, musical development, and their cultural heritage. The reason for this, as Swanwick [7] asserts, is "because (music) is a form of discourse as old as the human race, a medium in which ideas about ourselves and others are embodied in sonorous forms, ideas that may be simple or complex, obvious or enigmatic. And insight into these ideas—as into any significant idea—can be intrinsically rewarding."

Harnessing the contribution of music to children's education in primary schools remains largely dependent on teachers. While teachers would not require an extensive technical understanding of music theory or performance when teaching music appreciation, experimentation, and creativity, many primary teachers defer music instruction to their music-specialist colleagues, citing a lack of training, skill, confidence, or musical resources as reasons [8]. Hence, supporting general teachers to find new ways of including music within their classrooms and schools needs to be a priority [9].

It was against this background that a novel, school-based music intervention (named "Project Sparks") was designed and evaluated. This aimed to promote the musical, social, and emotional progression of young adults with disabilities by training them to become mentors for primary school children [10]. These young adults with disabilities, who all had undergone at least five years prior training in either singing or a musical instrument, will be referred to as "music mentors" throughout.

The initial aim of the intervention was to enlist the music mentors to improve children's access to participatory music creativity. However, given the absence of music-making in the schools where the intervention took place, the young adults with disabilities became, in a sense, peer mentors for the schoolteachers, providing them with practical tools to embed music in their classrooms without the need for extensive technical knowledge.

A growing body of literature has identified the value of peer mentoring and how it can be best used in education generally [11], as well as with music programmes [12]. Thus, the music mentors would model for teachers, new ways in which children can be engaged in music-making while also providing an additional resource that schools could call upon to assist with the promotion of music within the school curriculum.

This paper can be considered a quasi-feasibility study for the new type of intervention [13]. Details of the intervention are provided, but the main focus is to identify the impact of this approach that involved four schools in Northern Ireland and nearly 200 pupils aged around nine years. Of particular note is that the intervention would be tested with pupils from schools in a city with high levels of social deprivation. The children's musical aptitudes and attitudes toward disabled mentors were assessed before and after participating in the intervention as well as feedback from their parents but the main source of feedback came from the children's schoolteachers.

*Evaluating the Project's Impact*

This paper focusses on an evaluation of the project's impact, the aims of which were:

1. To determine the extent to which children's knowledge, aptitude, and confidence in music-making could be improved through their engagement with music mentors with disabilities.
2. To obtain children's views of the music mentors as educators and reactions of their parents to this approach.
3. To obtain primary school principals' and schoolteachers' views on the learning dynamic between their pupils and the music mentors.

## 2. Materials and Methods

### 2.1. Project Sparks

Two freelance music teachers created Project Sparks as a means of promoting the musical talents of young people with disabilities and their confidence as performers. The project initially involved a series of weekly training workshops, during which the music teachers mentored the musical skills of the young adults with disabilities (referred to as music mentors) allied with support from their peers during, but more especially, between workshops.

A range of training activities were employed, including role-plays, video-analysis of other instructors teaching, and practice workshops, in which they honed their skills with small groups of children from local primary schools. Within the sessions, the music

mentors taught these children some foundational concepts of music using singing and a range of tuned and untuned percussion instruments.

Initially, 22 young adults with disabilities joined the mentor-training sessions but ten dropped out, mainly due to the difficulty of the training and reluctance to perform and speak in front of others. Over the two years that the project ran, the remaining twelve mentors increased their competence as music tutors, as well as fostering a common "ownership" of their identity as disabled music mentors. Indeed, the young adults came to view their disabilities as a largely positive feature, enabling them to empathise with children who might face learning barriers [10].

*2.2. School-Based Music-Making*

After the project's pilot, further funding was obtained from a charitable foundation whose aim was to widen access to the arts, particularly among communities facing social inequalities. Four primary schools in a city with high levels of social deprivation agreed to participate in the feasibility study, involving eight classes and around 240 pupils aged around nine years. Across the four schools, the percentage of pupils entitled to free school meals (a commonly used indicator of social deprivation) ranged from 41–85% and had few pupils learning musical instruments (0–6%).

The twelve music mentors worked alongside the co-creators of the project (the first and second authors). Eight classes with their teachers took part in this phase of the project. Each class received 16 hours of music sessions over a four-week period, usually in the form of a four-hour workshop. However, a portion of the classes had to be modified due to COVID-19 restrictions. Some workshops were held on school premises, others in a performance space that was available to the project, and for one class, the sessions were delivered remotely via the internet.

The workshops aimed to develop pupils' foundational understanding of music. Various discovery-based activities were used to introduce pupils to "building-blocks" of music, namely rhythm, tempo, melody, and timbre. A range of readily available instruments were used such as drums, xylophones, djembes, and ukuleles. Pupils' exploration of each building-block was encouraged through the use of the instruments, their natural voices, and physical movement, to nurture their creative, aural, and motor-skills as they composed and appraised each other's ideas.

Moreover, the music activities were adjusted for each class to focus on certain academic skills that the teacher had identified as being most pertinent for his or her pupils. For example, in one class, pupils had difficulty understanding mathematical fractions, so the project staff devised simple rhythmic activities that illustrated fractions by making links to musical note values. With other classes, the mentors composed songs with the pupils to help pupils distinguish key literacy concepts such as adjectives, pronouns, verbs, and homophones. The mentors assisted the pupils in creating unique melodies and rhythms to fit around the words of the song.

Throughout the workshops, a learning mindset was promoted in which mistakes were not seen as a failure but rather as a learning opportunity with a strong focus on celebrating creative and novel musical experiences.

The role of the music mentors involved:

- Identifying the intended learning outcomes associated with each musical activity;
- Role-modelling successful musical compositions and performances;
- Facilitating small groups of pupils to encourage experimentations with the musical building-blocks they had learnt and fostering the sharing of ideas among the pupils;
- Providing pupils with praise and constructive feedback to improve their compositions;
- Answering questions about their musical interests and their disabilities during informal discussions at break and lunch times.

The pupils' teachers were present at all sessions and were encouraged to take on similar roles to the music mentors. Over the course of the workshops, the two project staff assisted the music mentors in reflecting upon the sessions, as well as using the mentors'

learning to plan for subsequent sessions. This process involved the music mentors analysing video-footage of their work during recent classes, as well as role-playing activities planned for future sessions. Weekly phone-calls were made by the project staff to the schoolteachers involved to discuss preparations for the next workshop session and any follow-up activities the teachers had undertaken with their class.

*2.3. Participants*

Disabled music mentors: Of the twelve disabled music mentors, seven were female and five male, with ages ranging from 19 to 28 years. They had a variety of impairments with some co-occurring: six were on the autism spectrum and six had learning disabilities, two had cerebral palsy; one spinal bifida and one Down syndrome. Six had attended special schools and six mainstream schools. Their chosen musical specialisms were singing ($n = 6$); drumming ($n = 5$); dance ($n = 3$); and song-writing ($n = 1$).

Children: The 171 children (51% male) who took part in the evaluation were in Year 6 of primary school and aged nine to ten years. (This represented around 85% of those in the classes with others missing due to absences when information was gathered).

Parents: The parents of 54 children (around 30% response) replied to a brief online questionnaire, but no demographic information was collected from them.

Teachers: Eight class teachers (five female and three male) from four different schools and two head teachers (one female, one male) were involved in the project and its evaluation. The length of time they had been teaching ranged from 6 to 27 years with a median of 12 years. One teacher had responsibility as the music co-ordinator for the school and two others had posts of responsibility for literacy and numeracy. Three teachers had taken music as a specialist subject in their education degree and five played a musical instrument, namely flute, piano, and cello.

*2.4. Evaluation of the Project*

The evaluation was designed to capture the experiences of pupils and teachers before and after their participation in the project. In addition, the reactions of parents were obtained after the project had been completed in their child's school. A mixed methods approach was chosen, using quantitative data obtained from specially developed rating scales with the pupils and parents, and with qualitative data through interviews with teachers and head teachers.

2.4.1. Children

The children's attitudes to and knowledge of music were assessed before and after participating in Project Sparks. They completed a computer-based, multiple-choice questionnaire that consisted of a series of items that had been used in a previous study and which reflected the objectives and content of the music sessions: a form of criterion-referenced assessment. Pupils completed the questionnaires individually in their classrooms with assistance from their teacher or classroom assistant if required. Three items related to the enjoyment of music and the answer options were: "never tried; don't enjoy, quite enjoy; really enjoy". Example items are shown in Table 1 (see results below).

**Table 1.** The means and (standard deviations) of summated scores for don't knows and correct responses on a seven item test of musical terms.

| Pre-Don't Know | Post-Don't Know | Pre-Correct | Post-Correct |
|:---:|:---:|:---:|:---:|
| 2.88 | 0.39 | 3.12 | 5.78 |
| (4.83) | (0.83) | (1.95) | (1.34) |

For items relating to music performance, the options were: "Don't know"; "Not important"; "Quite important"; and "Really important." These items included: "When Performing music with others, how important is it to play louder than everyone else?",

"How important is it to play in time with the pulse?", and "How important is it to blend in with the other instruments?"

Items relating to music listening or music creation also used a similar rating scale, relating to their usefulness: "Don't know"; "Not important"; "Quite important"; and "Really important." Example items were: "How useful is it to understand the story the composer is trying to tell?", "When creating music, how useful is it to use your imagination to paint your own story?", and "How useful is it to mix up the structure to try to confuse the listener?".

In the sections noted above, items were included that were reverse-coded in that a "not important" was indicative of a correct response.

Finally, knowledge of musical terms was tested by having the pupil select the correct answer from the alternatives presented. They were tempo; rhythm; pitch; timbre; dynamics; texture; and structure.

### 2.4.2. Parents

The children's parents were sent a text message about the project by the class teacher and sent a link to an online questionnaire to obtain their reactions to the project. The parents also rated a series of statements with four options provided. Sample items were: "How enthusiastic was your child about the project?", "Is your child any more or less interested in music since being on the project?", "Has your child shown an interest in learning to play a musical instrument?"

### 2.4.3. Teachers

Telephone interviews were conducted with the class teachers and with the headteachers from two of the participating schools. These were undertaken before and after classes had participated in the project by the fourth author who was independent from the project.

## 3. Results

### 3.1. Pupils' Perceptions of Music

Prior to the project, 30% of pupils indicated that they had never tried listening to new music and answering questions about it, but post-intervention, this percentage had dropped to 2.2%. Congruently, pre-intervention, 29.2% stated they "really enjoyed" listening to music, and post-intervention this had risen to 63.3%. Likewise, in response to the question: "How much do you enjoy experimenting with sounds?"; 18.8% stated they had "never tried' before the project, which fell to 1.4% afterward and the ratings for "really enjoy" rose from 37% to 64.7%. However, the children's ratings were not so different before and after to the question: "How much do you enjoy performing in front of an audience?"; 10.4% had "never tried' pre-intervention, and 2.2% post, while the "really enjoyed" ratings were 22.4% and 36.0%, respectively.

### 3.2. Performing Music

The ratings the pupils assigned to the ten items relating to music performance, listening, and creation were summated into one total score. (A principal components analysis identified that items loaded onto one factor that accounted for 52% of the variance with a Cronbach Alpha of 0.887, which is indicative of good internal reliability). Prior to the project, the children's mean score was 4.34 (SD 2.55), but this had risen significantly to 6.14 (SD 2.71) (Paired T-Tests, $p < 0.001$). Items for which the ratings "really important" had the largest increase were: "play in time to the pulse" (39.6% to 60.4%); to "not play quietly in case you make a mistake" (44.3% to 77.0%); and "blend in with other instruments" (22.9% to 43.2%).

### 3.3. Knowledge of Music

The children's knowledge of seven musical terms was summated into two scores. The number of items they had rated as "don't know" and the number of items they had

correctly answered. (The internal reliability of the seven items as indicated by Cronbach alpha was 0.861 on the pre-test questionnaires and 0.824 for the post-test). Table 1 shows the significant differences between the two sets of scores with the mean number of "don't knows" dropping to close to zero and the mean number of correct responses increasing (paired *t*-tests, $p < 0.001$).

### 3.4. Pupils Perceptions of Teachers with Disabilities

The children were asked to rate a single item: "It would be good to have a teacher with disabilities in my school". A five-point rating scale was provided ranging Big Yes, Yes, Maybe, No, Big No. Before the project, 43.6% chose Yes or Big Yes (with 45.3% selecting Maybe). After the project, the equivalent percentages were 62.6% Yes (31.7% Maybe).

A subset of pupils ($n = 116$) who took the project later were asked some additional questions to explore this issue further. Table 2 summarises their responses to items asked (note: the wording was changed in the questions asked after the project to the "leaders in the project"). For the first item, the options provided ranged "Not good"; "OK"; "Good"; "Very Good"; "Excellent". The last three items were scored out of 10.

**Table 2.** The means and (SDs) assigned to people/leaders with disabilities.

| Items | Before | After |
| --- | --- | --- |
| What kind of teacher would you imagine a disabled person being? Note: % rated excellent | 22.8% | 52.0% |
| How smart or stupid do you imagine disabled people to be? | 7.65 (1.95) | 9.16 (1.59) |
| How musically talented or not talented do you think disabled people are? | 7.87 (2.13) | 9.74 (0.66) |
| How much would you want a disabled person to teach you? | 6.48 (2.57) | 9.40 (1.33) |

As the table shows, there were large increases in all the ratings assigned to the music mentors with ceiling effects present as the scores after participating in the project came close to the maximum of 10 (Paired *t*-tests, $p < 0.001$).

### 3.5. Parents' Responses

Responses were received from 54 parents. In all, 80% of these respondents reported that their child had spoken to them about the project a great deal and 89% mentioned how their child was very enthusiastic about it. In all, 41% of parents reported that their child was "much more interested" in music following on from the project and another 41% considered them to be "more interested" (with 18% just the same). When asked: "Has your child shown an interest in learning to play a musical instrument?", 26% reported they were "much more interested" and 44% more interested with 28% just the same and 2% noting their child was less interested. In response to the question: "How important do you feel music is in your child's education?"; 57% rated it very important; 39% as important; and 4% as not really important.

Furthermore 85% of the children had talked to their parents about being taught by people with a disability and 80% of parents who responded felt it was "very important" for children to have this experience with the other 20% rating it "important". In all, 95% stated they would definitely recommend the project to other parents and schools.

Nearly half the parents added a comment generally praising the project. One commented:

"Fantastic project, that encouraged confidence and belief in my son and his abilities."

Another mother noted:

"My child just thought the sparks project was brilliant and talked about it and sang every time she came back from school. She loved the leaders and how they taught them songs and instruments and in her words were 'so amazing'. Well done to all involved as the songs and happiness sprinkled into our family."

*3.6. Teachers' Responses*

A thematic content analysis was undertaken by the fourth author from the transcripts of the audio-recorded interviews. The analysis was cross-checked with the two project leaders and further validated from the conversations they had had with the teachers during and after the workshop sessions.

All the class teachers and the two headteachers were enthusiastic about the project. Five themes were apparent in the analysis of their responses, and these are illustrated by quotations from individual respondents who are signified by the codes shown in brackets.

First, they spoke of the fun and excitement that the project engendered in the pupils and the confidence they had gained.

"There was such a real positive energy about the place all the time. No joke, from the minute the children arrived until they went home, every child got involved and every child was made to feel welcome and valued." (F1)

"We actually did a performance at a school assembly where they actually got to see the dance that they had put together There were several students that were so apprehensive to start, actually got up and performed so confidently." (C1)

"A concert was organised for us in the Waterside Theatre. The children were all invited to come and perform at it, on Saturday afternoon which they did. And the confidence on stage and everything was amazing to watch. And it is all thanks to the Sparks project. It is amazing." (F1)

"They pushed the children's self-esteem and confidence to a different level, a level that I have yet to see from other facilitators, natural, just a natural engagement with the children regardless of ability, or social background, or the varying sort of educational or social needs children may have, everybody was included." (HT)

This was apparent also in the pupils who took part online because of COVID restrictions:

"When I was talking to parents they said, the children came off the lessons and they were kind of buzzing about it and talking about the lessons . . . They would go up to their room with instruments, it was like a break away from everything that they could experiment and he said that they were just banging away up in the room at the drums." (PH)

Secondly the teachers commented on the range of musical experiences the project had covered.

"Their understanding now of the elements of music, of the texture, and melody and tempo and rhythm and all of that. They have a clear understanding of it and even now we try to play music in the class, they are able to keep the beat and find the beat themselves."(F1)

"They got the opportunity to learn all the different elements of music, you know, pitch, tempo, dynamics, rhythm. They learned it all through their little songs and rhymes. It also touched on aspects of maths which they really enjoyed. And I thought all the content in the actual project was very child friendly, and so enjoyable for all the children." (PG1)

A third theme was the cross-curricular content of the lessons.

"Their maths and their music skills; they were learning about prime numbers and homophones. They learned about resonant pulse and pitch and dynamics. And I then was able to bring that stuff back to school and deliver it then to the other children across the year group." (CHF)

"They learnt about prime numbers you know, they learned about fractions, it was all reinforced in a more fun way for them. And through wee rhymes and patterns and rhythms, through instruments, they were able to remember it more." (PG1)

"I would use it now, not necessarily just in music lessons. Like if we are doing literacy and numeracy, I can incorporate music in them now. And we were able to do a PE session based on music. It has changed how I look at music as a curricular subject." (F1)

A fourth theme related to the insights the teachers had gained into how music could be incorporated into their lesson plans.

"Teachers shy away from the idea of music because of the idea of making a mistake and it being heard and it being judged as teaching something wrong, and you want to make teachers aware that that is not the case." (F1)

"We wouldn't always give the time to music, but since going to Project Sparks and doing the teacher training in August, that's really motivated me. It's motivated other teachers too. Music's featuring again in our curriculum here, (after COVID restrictions were lifted) which is great. We weren't allowed to do it for so long, you know, so it's great that we can do it now, and we are." (CHF)

"It gives teachers an appreciation of possible music lessons and how they can link it to the curriculum and so on. So, not only is it just geared towards children, there's also an element of professional development as well within the programme." (HT)

A fifth theme revolved around the future and how projects like this can be sustained.

"There was a real sadness in the classroom that they weren't going on with this. They just wanted to keep it going because they were learning so much from it. So, maybe if we could expand on this and expand the learning from it as well." (F1)

"We have asked them the leaders to do in-service training with our staff because of the way they teach and the influence they had on my teaching, I kind of think this would be really good for other teachers to see." (F1)

"Music was an area for further development in the school development plan. And that in itself has adjusted the whole position of music and the teaching of music and learning of music in the school. I just wish that we had more funding just to get it right across the school." (CHF)

Few reservations about the project were expressed by the teachers, but one head teacher commented:

"It was a wee bit time consuming as regards to the weighting for the other areas of the curriculum like say literacy and numeracy and so on. The teachers found that they were playing a wee bit of catch-up with regard to all the other areas. If it had have been maybe done over more weeks; just had it have maybe sort of spread out it would have made it less pressurising." (HT)

## 4. Discussion

This project and its subsequent evaluation have a number of strengths despite its time-limited nature. The use of music mentors who had disabilities proved to be a success in terms of their engagement with pupils but also the non-threatening modelling it provided to the pupils' teachers on new ways of including music in their classrooms. The pupils' knowledge and appreciation of music showed significant improvements and their enthusiasm for music making had increased, which parents also confirmed. The teachers and head teachers identified five ways in which they had gained from Project Sparks that reinforced the potential value of music in the school curriculum. Of special note are the cross-curricular applications to priority subjects such as literacy and mathematics. Moreover, the project had focused on schools in areas of high social deprivation, which are thought to place less value on the arts and culture [14].

Resource limitations prevented a longer-term follow up with schools to document the extent to which teachers had continued promoting music within the school curriculum,

although informal contacts suggest that this has happened since. In particular, the project provided in-service music training to all school staff in one school, and another had funded repeat workshops for other classes and teachers in their school from the school budget. However, other schools were unable to do this due to the constraints on their budget and a reluctance to ask parents to make a financial contribution, which might be a possible option for schools in more affluent areas.

Paradoxically, the success of the project is no guarantee of effecting change around the place of music in the broader school curriculum when wider influences work against this occurring. These include the indicators of school performance that focus solely on literacy, numeracy, and sciences; the neglect of the arts in school inspections and the lack of support and training in music education provided to serving teachers especially [5].

Arguably, change may need to come from mobilizing teachers to advocate for a rebalancing of the primary school curriculum, which necessitates a change in their attitudes and knowledge about the place of music in their classroom practice. In that respect, projects such as Sparks provide a cost-effective means for facilitating changes in teachers' perceptions. But it is perhaps significant that such a change is better stimulated from outside the school community, for although primary schools may appoint personnel as music co-ordinators their role seemed to be quite circumscribed. It tends to prioritise the school's musical performances over a focus on children's musical development and/or the need to support colleagues of varying musical ability towards implementing substantive music education into their weekly schemes of work [9].

However, all the responsibility cannot be placed on teachers. An identified weakness of Project Sparks was that the carry-over from the pupils' appreciation of music into family life was not developed [15]. Of course, this may have been more likely to occur if greater efforts had been made to inform and involve parents in the project. Planned performances by the children of their musical accomplishments were not possible due to continuing COVID-19 restrictions, but this element should be included in future projects. Other options would include teachers sending photographs and video footage to the family Whatsapp group (or its equivalent) of their pupils' music-making. Nonetheless, the responses from parents do illustrate that school-based music interventions have the potential to positively influence the families' appreciation of music regardless of social status.

Finally, the leadership and expertise of the two project leaders should be acknowledged albeit that they worked through the music mentors by designing, structuring, and guiding their contributions through preparation, in vivo support, and reflections on what had gone well and needed improvement for future workshop sessions. In a sense, they provide an example of how music specialists might be better deployed in primary schools towards a training and support role for colleagues rather than focusing on musical performances with pupils [16]. Ironically, such personnel are at greater risk of becoming redundant when school budgets are cut.

Debates among educators about the place of music and the arts in general will no doubt continue. The value of innovative approaches such as Project Sparks is that they evidence what is possible and the outcomes achieved, not least the fun and enjoyment that the pupils and teachers experienced through their participation. In short, the controversy has to move beyond rhetoric into creating a new appreciation for music education in the school curriculum.

## 5. Conclusions

Although Project Sparks was a short, time-limited intervention, it did have a marked impact on the children's engagement in music making with increases in their knowledge and enjoyment of music. The use of music mentors who had disabilities demonstrated to the class teachers, new ways of including music in their classrooms. Moreover, the teachers and head teachers reported an increased appreciation of the value of music in children's education and particularly the cross-curricular applications to priority subjects such as

literacy and mathematics. Further research is needed to identify how innovative projects can be sustained and extended to many more teachers and schools.

**Author Contributions:** Conceptualization, E.M., E.C. and P.M.; methodology, E.M., E.C., P.M. and R.M.; formal analysis, R.M.; investigation, R.M.; resources, E.M. and E.C.; data curation, E.M.; writing—original draft preparation, R.M.; writing—review and editing, R.M., E.M., E.C. and P.M.; project administration, E.M.; funding acquisition, E.M. and E.C. All authors have read and agreed to the published version of the manuscript.

**Funding:** Project Sparks received funding from the Paul Hamlyn Foundation to undertake the work in schools and cover the costs of evaluating its impact.

**Institutional Review Board Statement:** Ethical review and approval were waived for this study, as under UK regulations, it was classed as an evaluation/audit of service delivery.

**Informed Consent Statement:** Informed consent was obtained from all participants involved in the study and assurances were given about the confidentiality of the information they provided.

**Data Availability Statement:** The data reported in this paper is available on reasonable requests to the corresponding author.

**Acknowledgments:** Our thanks to the schools and the head teachers for facilitating the collection of information from the pupils and parents.

**Conflicts of Interest:** The authors declare no conflict of interest.

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
