# Peer review of "Engaging Children in Music-Making: A Feasibility Study Using Disabled Musicians as Mentors in Primary Schools"

_education, doi:10.3390/educsci13010072_

Round 1

Reviewer 1 Report

Thank you for submitting an interesting, well written article on the engagement of children in music making. The participation of young people with disabilities makes this study unique. The article clearly describes the context in which the study has been conducted. The research design could have been described more clearly. What instruments have been used when? Plus, how has the data been analyzed? This applies especially to the qualitative data collection. By the way: I probably missed the meaning of F1, PH, HT, et cetera. 

Author Response

Reviewer #1

Response

Thank you for submitting an interesting well written article on the engagement of children in music making. The participation of young people with disabilities makes this study unique. The article clearly describes the context in which the study has been conducted.

Many thanks for your kind remarks. They are much appreciated.

The research design could have been described more clearly. What instruments have been used when?

We have elaborated on the research design and described more fully the instruments that were used. See section 2.4

Plus how has the data been analyzed? This applies especially to the qualitative data collection.

We have given more information on the analysis of the qualitative data. See section 2.4.1

By the way: I probably missed the meaning of F1, PH. HT.  et cetera.

We have highlighted the codes that were used to identify the respondents to the interviews.  See section 3.6

Reviewer 2 Report

The paper is correct. Nothing remarkable. My decision is to accept in present form.

Author Response

Reviewer #2

Response

The paper is correct. Nothing remarkable. My decision is to accept in present form.

Many thanks for your recommendation.